# On the modeling of amplitude-sensitive ESR detection using VCO-based ESR-on-a-chip detectors

Anh Chu[1], Benedikt Schlecker[1], Michal Kern[1], Justin L. Goodsell[2], Alexander Angerhofer[2], Klaus Lips[3], and Jens Anders[4]

[1]Institute of Smart Sensors, University of Stuttgart, Pfaffenwaldring 47, 70569 Stuttgart, Germany
[2]Department of Chemistry, University of Florida, Gainesville, FL32611-7200, USA
[3]Department Spins in Energy Materials and Quantum Information Science (ASPIN), Helmholtz-Zentrum Berlin für Materialien und Energie, Hahn-Meitner-Platz 1, 14109 Berlin, Germany
[4]University of Stuttgart, Institute of Smart Sensors and IQ$^{\mathrm{ST}}$ (Center for Integrated Quantum Science and Technology), Pfaffenwaldring 47, 70569 Stuttgart, Germany

**Correspondence:** Jens Anders (jens.anders@iis.uni-stuttgart.de)

**Abstract.** In this paper, we present an in-depth analysis of a voltage-controlled oscillator (VCO) based sensing method for electron spin resonance (ESR) spectroscopy, which greatly simplifies the experimental setup compared to conventional detection schemes. In contrast to our previous oscillator-based ESR detectors, where the ESR signal was encoded in the oscillation frequency, in the amplitude-sensitive method, the ESR signal is sensed as a change of the oscillation amplitude of the VCO. Therefore, using a VCO architecture with a built-in amplitude demodulation scheme, the experimental setup reduces to a single permanent magnet in combination with a few inexpensive electronic components. We present a theoretical analysis of the achievable limit of detection, which uses a perturbation theory based VCO-modeling for the signal and applies a stochastic averaging approach to obtain a closed-form expression for the noise floor. Additionally, the paper also introduces a numerical model suitable for simulating oscillator-based ESR experiments in a conventional circuit simulator environment. This model can, e.g., be used to optimize sensor performance early on in the design phase. Finally, all presented models are verified against measured results from a prototype VCO operating at 14 GHz inside a 0.5 T magnetic field.

## 1 Introduction

Electron spin resonance (ESR) is a very powerful spectroscopic method which is used extensively in a large variety of disciplines including chemistry, material science and the life sciences (Twahir et al., 2015, 2016; Azarkh et al., 2013; Qi et al., 2014; Qin and Warncke, 2015; Fehr et al., 2011, 2012). At its basis, ESR spectroscopy uses the spin of an electron as a very sensitive nanoscopic probe of its magnetic and electronic environment inside a molecule or a solid to provide important information, which are often difficult to obtain by other spectroscopy techniques. Since ESR detects exclusively paramagnetic species, it is ideally suited for the detection of free radicals, which are related to premature cell aging (Kopani et al., 2006), food degradation (Elias et al., 2009; Ottaviani et al., 2001) or for the detection of paramagnetic defects in semiconductor materials (Fehr et al., 2011). To overcome the problem of limited sensitivity in conventional ESR, miniaturized detectors have been suggested, which improve the achievable spin sensitivity thanks to their larger $B_{\mathrm{u}}$-field and, in this way, room temperature

spin sensitivities between $10^7 \, \mathrm{spins}/(\mathrm{G} \cdot \sqrt{\mathrm{Hz}})$ and $10^9 \, \mathrm{spins}/(\mathrm{G} \cdot \sqrt{\mathrm{Hz}})$ at various $B_0$-field strengths have been reported in the literature, cf. Anders et al. (2012a); Twig et al. (2013); Gualco et al. (2014); Matheoud et al. (2017, 2018); Dayan et al. (2018); Abhyankar et al. (2020); Zhang and Niknejad (2021). Apart from the poor sensitivity associated with inductive ESR detectors, conventional ESR setups also suffer from a relatively large complexity. As a partial solution to this problem, an oscillator-based ESR detection method was presented by Anders et al. (2012a) and Yalcin and Boero (2008) which detects the ESR effect by monitoring the sample-induced inductance variation as a change in the oscillation frequency. By using integrated LC tank oscillators, this approach removes the need for expensive external $B_1$-field sources and also benefits from the great scaling potential of modern nanometer-scaled CMOS technologies and their very high maximum operating frequencies. Exploiting these advantages and utilizing a $45 \, \mathrm{\mu m}$ detection coil inside an LC tank oscillator operating around $146 \, \mathrm{GHz}$, the design presented by Matheoud et al. (2017) achieves a spin sensitivity of about $2 \times 10^7 \, \mathrm{spins}/(\mathrm{G} \cdot \sqrt{\mathrm{Hz}})$. The oscillator-based detection concept was then extended to the use of voltage-controlled oscillators (VCOs) by Handwerker et al. (2016), which allows for a great simplification of the experimental setup, thereby, for the first time, enabling the design of battery-operated, portable ESR spectrometers. Such portable ESR spectrometers can potentially have a very large impact on (emerging) disciplines such as the analysis of irradiated food (Chauhan et al., 2009), the study of wine oxidation (Elias et al., 2009), the prevention of the formation of free radicals in vegetable oils (Ottaviani et al., 2001), on-site radiation dosimetry (Romanyukha et al., 2014), point-of-care transcutaneous oxygen monitoring (Wolfson et al., 2014; Cristea et al., 2021) or measurements of skin antioxidant capability (Haag et al., 2011).

While in the reports by Anders et al. (2012a), Matheoud et al. (2017) and Yalcin and Boero (2008) only the frequency-sensitive detection option of an LC tank oscillator was discussed, a second mode of detection is available in oscillator-based ESR detectors because the oscillation amplitude is also affected by the ESR signal. This concept was originally published by Chu et al. (2018) using CMOS LC-tank VCO and by Matheoud et al. (2018) using a high electron mobility transistor based LC Colpitts oscillator. In both of those reports, amplitude-sensitive detection is mentioned beside frequency-sensitive detection and sensitivity calculations are performed only for the latter. In this paper, we will extend the state-of-the-art by providing both analytical and numerical models for the amplitude-sensitive detection mode. Using our analysis, we show that the amplitude and frequency-sensitive detection modes display the same theoretically achievable spin sensitivity but with the potential for a simplified experimental setup for the amplitude-sensitive detection mode. These simplifications can, in turn, be used for further reductions in the size and cost of future generations of portable ESR spectrometers.

The paper is organized as follows. In section 2, we will explain the experimental setup of an amplitude-sensitive VCO-based ESR experiment. In section 3 and 4 we then derive analytical expressions for the ESR-induced amplitude variations in an LC tank oscillator before we also provide analytical expressions for the amplitude noise of LC tank VCOs in section 5 to estimate the achievable limit of detection (LOD) in section 6. Next, in section 7 we provide a model suitable for simulating ESR spectroscopy experiments in conventional circuit simulators. Then, in sections 8 and 9, we compare the analytical model against these circuit simulations and validate all models using measured results from a VCO prototype operating around $14 \, \mathrm{GHz}$ in a $0.5 \, \mathrm{T}$ magnetic field. The paper is concluded with a short discussion and an outlook on future work in section 10.

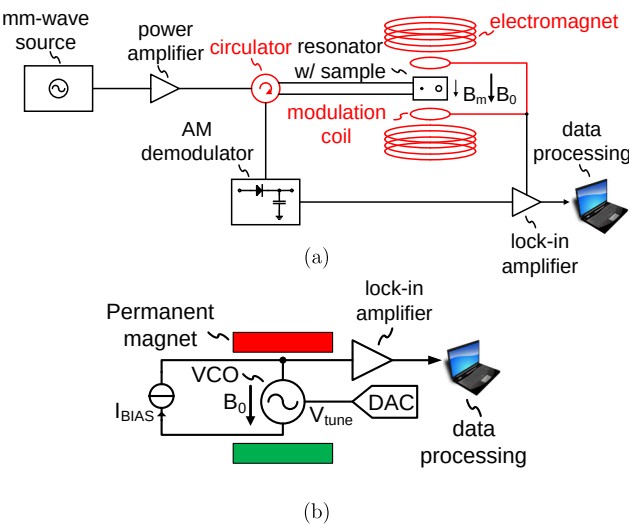

**Figure 1.** (a) Conventional ESR detection setup and (b) Proposed ESR detection scheme, which measures the ESR effect as a change in the oscillation amplitude of an integrated LC tank VCO.

## 2 Performing amplitude-sensitive ESR experiments using LC tank VCOs

A conventional setup for ESR experiments is shown in Fig. 1a. The ESR sample is placed inside a microwave resonator which is situated inside a variable field magnet. An ESR experiment is performed by irradiating the sample with a microwave at a constant frequency through a circulator and monitoring the reflected power. The external magnetic field $B_0$ is swept through the resonance condition. In order to improve the achievable sensitivity, frequently lock-in detection is introduced by modulating the static magnetic field with an AC magnetic field with amplitude $B_m$ using a pair of modulation coils. The building blocks highlighted in red in Fig. 1a are those that prevent a miniaturization of the experimental setup (electromagnet), an energy efficient operation (electromagnet and modulation coils) and an integration into CMOS technology of the spectrometer electronics (circulator). In contrast, in the amplitude-sensitive detection scheme incorporating VCOs shown in Fig. 1b, all required electronic components can be easily integrated into CMOS technology and the power-hungry and bulky electromagnet is replaced by a permanent magnet. The replacement of the variable field by a permanent magnet is possible because in the proposed setup, an ESR spectrum is recorded at a fixed static magnetic field $B_0$ while sweeping the frequency of the excitation signal (i.e. the magnetic field produced by the coil of the integrated LC tank oscillator) in and out of resonance to induce the ESR transition. Using a VCO, this frequency sweep can conveniently be carried out by applying a voltage ramp to the VCO control voltage using a digital-to-analog converter (DAC). The VCO control voltage both defines the new excitation frequency, and, at the same time, tunes the LC tank inside the VCO to this frequency. This is because, in a VCO, the oscillation frequency and the resonance frequency of the LC tank are identical at all times. This is in contrast to a conventional resonator-based scheme, in which the resonance frequency and the excitation frequency can be independently defined. Moreover, the same DAC output signal can be used to produce a frequency modulation at every sweep point, which allows to replace the field modulation using

external modulation coils by a much more power saving frequency modulation with the same positive effect on the achievable SNR when using subsequent lock-in detection. At this point, it is important to note that the aforementioned simplifications of the experimental setup were already achieved using the frequency-sensitive VCO-based detection setup presented by Handwerker et al. (2016). However, the amplitude-sensitive ESR setup of Fig. 1b provides the additional advantage of an implicit demodulation of the ESR signal. More specifically, when using current biasing for the LC tank oscillator according to Fig. 2a,

the voltage at the center tap of the differential tank inductor (node $\boxed{X}$ in the figure) contains a demodulated version of the oscillation amplitude (Kinget, 2006). This implicit AM demodulation feature of the VCO not only removes the necessity of an external AM demodulation block but also minimizes the number of high frequency components because the lock-in amplifier can directly be connected to the inductor center tap voltage, cf. Fig. 1b. This further simplifies the experimental setup compared to the frequency-sensitive detection used by Handwerker et al. (2016). In the approach of Handwerker et al. (2016), the VCO

output signal first had to be processed by a chain of frequency dividers to allow for simplified analog-to-digital conversion and subsequent frequency demodulation by a digital phase-locked loop. In the proposed amplitude-sensitive ESR setup, an implicit AM demodulator inside the VCO is used, resulting in the very simple experimental setup of Fig. 1b.

## 3   Deterministic model of the amplitude and frequency of an LC tank VCO

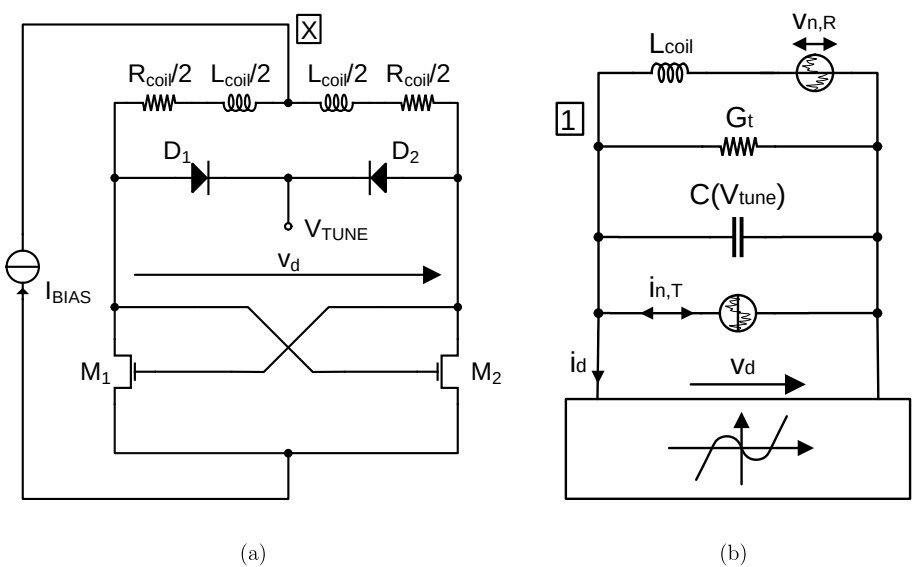

(a)                                        (b)

**Figure 2.** (a) Schematic of the CMOS LC tank VCO and (b) equivalent circuit modeling the current starved cross-coupled transistor pair as a static third order nonlinearity.

In order to be able to derive an analytical expression for the ESR induced amplitude changes in the oscillation amplitude of

90 an LC tank VCO according to Fig. 2a, we will first derive closed-form expressions for the oscillation amplitude and frequency. As the starting point for our analysis we will use the equivalent electrical model of the schematic of Fig. 2a shown in Fig. 2b,

where $G_t = 1/(R_{coil} \cdot Q^2_{coil})$ is the equivalent tank conductance, $Q_{coil}$ being the coil quality factor, and $v_{n,R}$ and $i_{n,T}$ are noise sources modeling the noise generated in the coil resistance and the cross-coupled transistor pair, respectively. For the following deterministic analysis, the noise source will be set to zero and they will only be considered for the noisy case discussed in section 5. To obtain an I/V-characteristic of the static nonlinearity of Fig. 2b, which models the cross-coupled transistor pair, we have followed the approach proposed by Anders et al. (2012c), resulting in:

$$i_d \approx -\frac{G_{m0}}{2} v_d + \frac{G^3_{m0}}{16\, I^2_{BIAS}} v^3_d \tag{1}$$

where $G_{m0} = \sqrt{\beta I_{BIAS}/n}$ is the gate transconductance (Enz and Vittoz, 2006) of a single transistor in the cross-coupled differential pair for $v_d = 0$, $n \approx 1.3$ is the transistor slope factor (Enz and Vittoz, 2006) and $I_{BIAS}$ is the oscillator bias current. Then, using the differential tank voltage as state variable $x = v_d$ and applying Kirchhoff's current law to node $\boxed{1}$, we obtain the following ordinary differential equation describing the oscillator behavior:

$$\ddot{x} + \omega^2_{LC} x = -\varepsilon \frac{1}{C} \left( \frac{1}{\varepsilon} \left[ G_t - \frac{G_{m0}}{2} \right] + \frac{x^2}{I^2_{BIAS}} \right) \dot{x} \tag{2}$$

where $\varepsilon = 3\, G^3_{m0}/\left(16\, n^2\right)$, $G_{m0}$ being the gate transconductance and $n \approx 1.3$ the slope factor, $G_t$ is the equivalent tank conductance, $I_{BIAS}$ is the bias current and $\omega_{LC} = 1/\sqrt{L_{coil} C}$ is the resonance frequency of the LC tank. Starting from eq. (2) we can use the so-called Lindtstedt method (Jordan and Smith, 2007) to obtain first order estimates of the oscillation amplitude and frequency according to:

$$A_{osc,0} = 4 \sqrt{\frac{2}{3} \frac{n\, I_{BIAS}}{G_{m0}}} \sqrt{1 - \frac{1}{\alpha_{od}}} \tag{3a}$$

$$\omega_{osc} = \omega_{LC} \left( 1 - \frac{(\alpha_{od} - 1)^2}{16\, Q^2_{coil}} \right), \tag{3b}$$

where $\alpha_{od} = G_{m0}/(2\, G_t)$ is the overdrive parameter, which needs to be chosen greater than one to ensure a stable oscillation and all other parameters are defined as above.

## 4 ESR-induced amplitude shifts

As explained by Yalcin and Boero (2008), the effect of ESR on the spin ensemble can be modeled by means of a complex susceptibility $\chi = \chi' - j\chi''$ according to:

$$\chi' = \frac{\Delta\omega\, T^2_2}{1 + (T_2\, \Delta\omega)^2 + (\gamma\, B_1)^2\, T_1\, T_2} \omega_L\, \chi_0 \tag{4a}$$

$$\chi'' = -\frac{T_2}{1 + (T_2\, \Delta\omega)^2 + (\gamma\, B_1)^2\, T_1\, T_2} \omega_L\, \chi_0, \tag{4b}$$

where $\Delta\omega = \omega_{osc} - \omega_L$, $\omega_{osc}$ being the oscillation frequency and $\omega_L = -\gamma B_0$ being the electron Larmor frequency (Schweiger and Jeschke, 2001), where $\gamma$ and $B_0$ are the gyromagnetic ratio of electrons[1] and the applied static magnetic field strength, $T_1$

---
[1]For a free electron, we have $\gamma/(2\pi) \approx$ -28.025 GHz/T.

and $T_2$ are the longitudinal and transverse relaxation times, respectively, and $\chi_0$ is the static electron susceptibility. Using the complex susceptibility, the effective tank coil impedance in the presence of a resonant electron spin ensemble can be written as $Z_\chi = j\omega_{\mathrm{osc}} L_{\mathrm{coil}}(1 + \eta\chi)$, where $\eta$ is the so-called filling factor (Yalcin and Boero, 2008), which indicates how much of the sensitive coil volume is effectively filled by the ESR active material. Therefore, the effective coil inductance and coil resistance in the presence of ESR, $L_\chi$ and $R_\chi$, can be written according to $L_\chi = L_{\mathrm{coil}}(1 + \eta\chi')$ and $R_\chi = R_{\mathrm{coil}}(1 + Q_{\mathrm{coil}}\eta\chi'')$, respectively, where $Q_{\mathrm{coil}}$ is the coil quality factor. In order to obtain the oscillation voltage and frequency including the effect of ESR, we can start from eq. (3) and replace the original coil inductance and resistance (i.e. in the absence of ESR), $L_{\mathrm{coil}}$ and $R_{\mathrm{coil}}$, by their effective values in the presence of ESR, i.e. $R_\chi$ and $L_\chi$, respectively. Since in this paper, we are only interested in the ESR-induced amplitude changes, in the following, we will only consider the effect of ESR on the oscillation amplitude described by eq. (3a), yielding:

$$A_{\mathrm{osc},\chi} \approx 4\sqrt{\frac{2}{3}}\frac{n\,I_{\mathrm{BIAS}}}{G_{\mathrm{m0}}}\sqrt{1 - \frac{2G_{\mathrm{t},\chi}}{G_{\mathrm{m0}}}}, \tag{5}$$

where $G_{\mathrm{t},\chi}$ is the equivalent tank conductance in the presence of ESR. The equivalent tank conductance depends on both the coil resistance $R_{\mathrm{coil}}$ and – via the coil quality factor $Q_{\mathrm{coil}}$ – on the coil inductance $L_{\mathrm{coil}}$, according to $G_{\mathrm{t},\chi} = C\,R_{\mathrm{coil},\chi}/L_{\mathrm{coil},\chi}$, eq. (5) can be rewritten according to:

$$A_{\mathrm{osc},\chi} \approx 4\sqrt{\frac{2}{3}}\frac{n\,I_{\mathrm{BIAS}}}{G_{\mathrm{m0}}}\sqrt{1 - \frac{2C\,R_{\mathrm{coil}}(1 + Q_{\mathrm{coil}}\eta\chi'')}{G_{\mathrm{m0}}\,L_{\mathrm{coil}}(1 + \eta\chi')}}. \tag{6}$$

Eq. (6) can be further simplified by noting that the ESR-induced inductance changes are much smaller than the original coil inductance, that is $\eta\chi' \ll 1$. Consequently, the term $1/(1 + \eta\chi')$ can be developed into a Taylor series in $\eta$ around $\eta = 0$, which can be stopped after the linear term and eq. (6) simplifies to:

$$A_{\mathrm{osc},\chi} \approx 4\sqrt{\frac{2}{3}}\frac{n\,I_{\mathrm{BIAS}}}{G_{\mathrm{m0}}}\sqrt{1 - \frac{1}{\alpha_{\mathrm{od}}}(1 + \eta[Q_{\mathrm{coil}}\chi'' - \chi'])}, \tag{7}$$

where we have neglected the quadratic term in $\eta^2$, which originates from the product $(1 + Q_{\mathrm{coil}}\eta\chi'') \cdot (1 - \eta\chi')$. To arrive at a closed-form expression for the ESR-induced amplitude changes, we can further develop the right hand side of eq. (7) into a first order Taylors series in $\eta$ around $\eta = 0$. Then, the ESR-induced amplitude change $\Delta A_{\mathrm{osc},\chi} \triangleq A_{\mathrm{osc},\chi} - A_{\mathrm{osc},0}$ can be written as:

$$\Delta A_{\mathrm{osc},\chi} = \underbrace{4\sqrt{\frac{2}{3}}\frac{n\,I_{\mathrm{BIAS}}}{G_{\mathrm{m0}}}\sqrt{1 - \frac{1}{\alpha_{\mathrm{od}}}}}_{A_{\mathrm{osc},0}} \cdot \frac{\eta(Q_{\mathrm{coil}}\chi'' - \chi')}{2(\alpha_{\mathrm{od}} - 1)}. \tag{8}$$

According to eq. (8), the ESR-induced amplitude change depends on both the real part of the complex susceptibility, $\chi'$, and its imaginary part, $\chi''$. However, for moderate coil quality factors with $Q_{\mathrm{coil}} \gg 1$, the term $Q_{\mathrm{coil}}\chi''$ largely dominates, and the ESR induced amplitude changes mostly depend on the imaginary part of the complex susceptibility according to:

$$\Delta A_{\mathrm{osc},\chi} \approx A_{\mathrm{osc},0}\frac{\eta \cdot Q_{\mathrm{coil}}\chi''}{2(\alpha_{\mathrm{od}} - 1)}. \tag{9}$$

## 5 Model of amplitude noise in LC tank VCOs

Due to the great importance of timing uncertainties on the overall system of modern communication systems, oscillator phase noise is probably one of the most-discussed topics in RF circuit theory and a wide variety of models with different degrees of complexity exist in the literature ranging from simple linear time-invariant over linear time-varying to more complicated nonlinear models (Kaertner, 1990; Hajimiri and Lee, 1998; Demir, 2002; Nallatamby et al., 2003; Magierowski and Zukotynski, 2004; Andreani et al., 2005; Sancho et al., 2007; Jahanbakht and Farzaneh, 2010; Murphy et al., 2010). One fundamental problem associated with oscillator noise modeling is related to the fact that an oscillator is a nonlinear system away from thermal equilibrium. This leads to a situation where even the most sophisticated models available today, which rely on modeling using stochastic differential equations (SDEs), can be considered as heuristics only. This is because the Langevin approach of introducing additional additive noise sources into the system in general fails for nonlinear dynamical systems, leading to physical inconsistencies (Thiessen and Mathis, 2010). Here, the problem essentially arises from the coupling between the different moments of the stochastic process described by the SDE, which results in a situation where the stochastically averaged SDE is in general not identical to the deterministic system to which the noise sources have been added. Therefore, due to the heuristic nature of even the most advanced models proposed in the literature, a validation against simulations and – even more importantly – against measured data is crucial. While for oscillator phase noise such experimentally verified heuristic models exist, the field of oscillator amplitude noise is by far less explored and there is only a very small set of papers which deal with this topic typically as a side note without experimental verification (Magierowski and Zukotynski, 2004; Jahanbakht and Farzaneh, 2010). This is mostly because the oscillator amplitude noise is of negligible importance for the resulting timing uncertainty and is therefore neglected in analysis papers focusing on oscillator applications in RF systems. However, in sensor systems, which use the oscillator to measure a physical quantity as an amplitude change of the oscillator output voltage, the oscillator amplitude noise determines the achievable limit of detection and its accurate modeling is of utmost importance. This includes the amplitude-sensitive ESR detection mode discussed in this paper but also eddy-current crack detection sensors for nondestructive testing (NDT) (García-Martín et al., 2011). Due to the lack of existing models on oscillator amplitude noise in the literature, in this section, we will present a model based on the stochastic averaging method proposed by Stratonovich (1963), which takes into account the nonlinearity of the oscillator but still produces closed-form expression for the autocorrelation and power spectral density of the resulting amplitude noise process.

We have already applied the stochastic averaging method to an LC tank oscillator to obtain analytical expressions for the phase and frequency noise of such circuits and verified its accuracy using measured data (Anders et al., 2012b). Following the method outlined by Anders et al. (2012b) and Anders (2011), one can derive the following SDE governing the behavior of the amplitude noise, $\delta A$, of the current starved LC tank oscillator of Fig. 2:

$$\delta \dot{A}(t) = \underbrace{-(\alpha_{\mathrm{od}} - 1)\frac{R_{\mathrm{coil}}}{L_{\mathrm{coil}}}}_{\triangleq \lambda} \delta A - \omega_{\mathrm{LC}}\, v_{\mathrm{n,R}} \sin\left(\omega_{\mathrm{osc}}t + \varphi_0\right) + \frac{1}{C}\, i_{\mathrm{n,T}} \sin\left(\omega_{\mathrm{osc}}t + \varphi_0\right), \tag{10}$$

where $v_{n,R}$ and $i_{n,T}$ are the noise sources modeling the noise introduced by the coil resistance and the active cross-coupled transistor pair in Fig. 2. Then, introducing the stochastic process $\xi_n(t) = \omega_{LC} v_n(t) - 1/C\, i_n(t)$ into eq. (10), one obtains:

$$\delta\dot{A}(t) = -\lambda\delta A(t) - \xi_n(t)\sin\left(\omega_{osc}t + \phi_0\right). \tag{11}$$

Eq. (11) defines a time-dependent Ornstein-Uhlenbeck process (Gardiner, 2009) and its solution, assuming a vanishing initial condition at $t \mapsto -\infty$, is therefore given by:

$$\delta A(t) = -\int\limits_{-\infty}^{t} \exp\left(-\lambda\left[t - t'\right]\right)\xi_n(t')\cos\left(\omega_{osc}t' + \varphi_0\right))\mathrm{d}t'. \tag{12}$$

Assuming that $v_{n,R}$ and $i_{n,T}$ are Gaussian random processes with zero mean, $\delta A$ will also be Gaussian with vanishing mean. Consequently, the autocorrelation of $\delta A$, $R_{\delta A\delta A}(t,\tau)$, is sufficient to completely characterize the statistics of the amplitude noise. This autocorrelation is given by:

$$R_{\delta A\delta A}(t,\tau) = \frac{1}{2}\int\limits_{-\infty}^{t}\int\limits_{-\infty}^{t+\tau} \exp\left(-\lambda\left[2t + \tau - t' - t''\right]\right)R_{\xi\xi}(t',t'')\cos(\omega_{osc}[t'' - t'])\mathrm{d}t'\mathrm{d}t'', \tag{13}$$

where it was further assumed that the initial phase $\varphi_0$ is a random variable uniformly distributed in the interval $[0, 2\pi]$. The double integral of eq. (13) can be solved in closed form if one assumes that $\xi_n$ is white, i.e. $R_{\xi\xi}(t_1, t_1+\tau) = R_{\xi\xi}(\tau) = \alpha_n^2\delta(\tau)$, using a variable transformation, cf. Stratonovich (1963), according to $\sigma = t' - t'' + \tau$ and $s = (t' + t'')/2$, yielding:

$$R_{\delta A\delta A}(t,\tau) = 2\left(\frac{\alpha_n}{2\lambda}\right)^2\exp\left(-\lambda|\tau|\right)\cos\left(\omega_{osc}t\right), \tag{14}$$

where the noise scaling coefficient $\alpha_n^2$ of the process $\xi_n$ was calculated by Anders et al. (2012b) as

$$\alpha_n^2 = kT R_{coil}\omega_{LC}^2\left(1 + \alpha_{od}\gamma_{nD}\right) \tag{15}$$

where $k$ is Boltzmann's constant, $T$ is absolute temperature, and $\gamma_{nD}$ is the thermal noise excess factor (Enz and Vittoz, 2006) of a MOSFET transistor with $\gamma_{nD} = 2/3n \approx 1$ for a transistor in strong inversion and saturation. The corresponding power spectral density, which is centered around $\omega_{osc}$, is then given by:

$$S_{\delta A\delta A}(\Delta\omega) = \frac{1 + \alpha_{od}\gamma_{nD}}{(\alpha_{od} - 1)^2}Q_{coil}^2 kT R_{coil}\frac{1}{1 + \left(\frac{\Delta\omega}{\omega_c}\right)^2}, \tag{16}$$

where $\Delta\omega = \omega - \omega_{osc}$ and $\omega_c = \lambda$ is the resulting corner frequency.

## 6  Limit of detection

In this section, the results from the previous two sections will be combined to obtain the limit of detection (LOD), i.e. the minimum number of spins detectable with an SNR of three in one second of measuring time, of an amplitude-sensitive VCO-based ESR detector. In order to make the results comparable with previously published resonator-based and frequency-sensitive

oscillator-based ESR experiments, we will introduce the standard ESR terminology into the LOD expression. To this end, one can recast the result of eq. (9) by noting that the oscillation amplitude and the $B_1$-field, i.e. the magnetic field produced by the oscillation current in the tank inductor, are related according to $A_{\mathrm{osc},0} \approx \omega_{\mathrm{osc}} L_{\mathrm{coil}} \hat{I}_{\mathrm{coil}} = \omega_{\mathrm{osc}} (2 B_1 B_{\mathrm{u}} V_{\mathrm{det}})/\mu_0$, where $B_{\mathrm{u}}$ is the unitary magnetic field of the detection coil, $V_{\mathrm{det}}$ is the sensitive detector volume and $\mu_0$ is the vacuum permeability[2]. Then, substituting $A_{\mathrm{osc},0}$ in eq. (9) by the above expression, we find:

$$\Delta A_{\mathrm{osc}} = \frac{B_1 B_{\mathrm{u}} V_{\mathrm{det}} \omega_{\mathrm{osc}} Q_{\mathrm{coil}}}{\mu_0 (\alpha_{\mathrm{od}} - 1)} \cdot \eta \chi''(\Delta\omega), \tag{17}$$

where we have used the notation $\chi''(\Delta\omega)$ to emphasize the fact that the imaginary part of the complex susceptibility is a function of the frequency offset $\Delta\omega = \omega_{\mathrm{osc}} - \omega_{\mathrm{L}}$ between the oscillation frequency $\omega_{\mathrm{osc}}$ and the Larmor frequency $\omega_{\mathrm{L}} = -\gamma B_0$ of the electron spins at the static magnetic field strength $B_0$. Using the analytical expression for the oscillator amplitude noise of eq. (16) evaluated at $\Delta\omega = 0$ and assuming a detection bandwidth of $f_{\mathrm{BW}}$, we can write the SNR of amplitude-sensitive ESR experiments as a function of both $\omega_{\mathrm{osc}}$ and $B_1$ according to:

$$\mathrm{SNR}(\omega_{\mathrm{osc}}, B_1) = \frac{B_1 B_{\mathrm{u}} V_{\mathrm{det}} \omega_{\mathrm{osc}} \eta \chi''(\Delta\omega)}{\mu_0 \sqrt{(1 + \alpha_{\mathrm{od}} \gamma_{\mathrm{nD}}) kT R_{\mathrm{coil}} f_{\mathrm{BW}}}}. \tag{18}$$

To find the maximum SNR, we can substitute the imaginary part of the complex susceptibility by the term including saturation in eq. (4b), then take the partial derivatives of eq. (18) with respect to $B_1$ and $\omega_{\mathrm{osc}}$, equate them to zero, and find the following optimum $B_1$ field strength and oscillation frequency $\omega_{\mathrm{osc}}$, respectively:

$$B_{1,\mathrm{opt}} \approx \sqrt{\frac{1}{T_1 T_2}} \cdot \frac{1}{\gamma} \tag{19}$$

$$\omega_{\mathrm{osc,opt}} \approx \omega_{\mathrm{L}}. \tag{20}$$

Substituting these values for $B_1$ and $\omega_{\mathrm{osc}}$ into eq. (18) we find the following expression for the maximally achievable SNR:

$$\mathrm{SNR}_{\mathrm{opt}} \approx \frac{B_{\mathrm{u}} \chi_0 \eta V_{\mathrm{det}} \omega_{\mathrm{L}}^2}{2\gamma \mu_0 \sqrt{(1 + \alpha_{\mathrm{od}} \gamma_{\mathrm{nD}}) kT R_{\mathrm{coil}} f_{\mathrm{BW}}}} \cdot \sqrt{\frac{T_2}{T_1}} \tag{21}$$

Since the longitudinal relaxation time $T_1$ is always greater than or equal to half the transversal relaxation time $T_2$, i.e. $2T_1 \geq T_2$, the SNR of eq. (21) is maximized for $2T_1 = T_2$. Using the optimum achievable SNR in eq. (21), we can define the spin sensitivity $N_{\mathrm{min}}$ according to:

$$N_{\mathrm{min}} = \frac{3 N_{\mathrm{spins}}}{\mathrm{SNR}_{\mathrm{opt}}(\eta = 1, f_{\mathrm{BW}} = 1\,\mathrm{Hz})} \tag{22}$$

where $N_{\mathrm{spins}}$ is the number of spins in the sample that produces the optimum SNR, $\mathrm{SNR}_{\mathrm{opt}}$, for a filling factor of $\eta = 1$ with a detection bandwidth of $f_{\mathrm{BW}} = 1\,\mathrm{Hz}$. Noting that the static electron susceptibility $\chi_0$ can be expressed as $\chi_0 = \mu_0 N \gamma^2 \hbar^2 /(4kT)$ (Yalcin and Boero, 2008), where $N$ is the spin density of the sample, $\hbar$ is the reduced Planck constant,

---

[2]The coil inductance $L_{\mathrm{coil}}$ can be computed from the unitary field according to $L_{\mathrm{coil}} = 1/\mu_0 \cdot \int |\boldsymbol{B}_{\mathrm{u}}|^2 dV \approx 1/\mu_0 \cdot B_{\mathrm{u}}^2 \cdot V_{\mathrm{det}}$. Moreover, according to standard ESR terminology, the $B_1$-field is the circularly polarized field produced by the coil current in resonance with the spin ensemble, i.e. $B_1 \approx \mu_0/d_{\mathrm{coil}} \cdot \hat{I}_{\mathrm{coil}}/2 = B_{\mathrm{u}} \cdot \hat{I}_{\mathrm{coil}}/2$.

$\mu_0$ is the vacuum permeability, $\gamma$ is the gyromagnetic ratio, $k$ is Boltzmann's constant and $T$ is absolute temperature, the theoretical spin sensitivity of the proposed amplitude-sensitive ESR detection method can be expressed as:

$$N_{\min} = 12\sqrt{2}\,\frac{k^{3/2}T^{3/2}\sqrt{(1+\alpha_{\mathrm{od}}\,\gamma_{\mathrm{nD}})\,R_{\mathrm{coil}}}}{\hbar^2\,\gamma^3\,B_{\mathrm{u}}\,B_0^2}. \tag{23}$$

Since the spin sensitivity given by eq. (23) is identical to the one given by Yalcin and Boero (2008) (except for the factor $\sqrt{2(1+\alpha_{\mathrm{od}}\,\gamma_{\mathrm{nD}})}$ which accounts for the different condition of $T_1 = T_2$ used by Yalcin and Boero (2008), and the noise originating in the cross-coupled transistor pair which was not considered there), the theoretically achievable spin sensitivity of an LC tank oscillator is identical for the amplitude and the frequency-sensitive detection modes and also identical to that of a conventional resonator-based ESR detector.

## 7   Simulating ESR experiments using circuit simulators

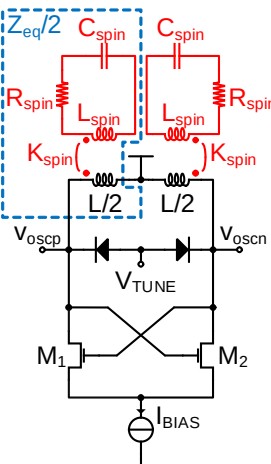

**Figure 3.** Model suitable to simulate VCO-based continuous-wave ESR experiments in a conventional circuit simulator. The effect of the spin ensemble on the oscillator is modeled by an inductive coupling between an RLC cirucit (model for the spin ensemble) that couples inductively into the tank inductance of the oscillator circuit, cf. circuitry inside the blue dashed line in the figure.

To design CMOS VCO-based ESR detectors with optimum performance, it is important to be able to accurately simulate the achievable sensitivity including all transistor nonidealities. To this end, in this section, we will provide a model which is suitable for simulating the effect of ESR on the frequency and the amplitude of CMOS LC tank VCOs in conventional circuit simulators. The utilized model was first proposed by Boero (2000) in the context of conventional resonator based nuclear

magnetic resonance (NMR) experiments but can also be applied to continuous-wave VCO-based ESR experiments in the nonsaturated case, i.e. for $(\gamma B_1)^2 T_1 T_2 \ll 1$, where the expressions for the complex magnetic susceptibility of eq. (4) simplify

to:

$$\chi' \approx \frac{\Delta\omega\, T_2^2}{1 + (T_2\,\Delta\omega)^2}\,\omega_L\,\chi_0 \tag{24a}$$

$$\chi'' \approx -\frac{T_2}{1 + (T_2\,\Delta\omega)^2}\,\omega_L\,\chi_0. \tag{24b}$$

Then, by comparing the impedance of a coil filled with spins, $Z_\chi = L_{\mathrm{coil}}\,(1 + \eta\chi)$, whose susceptibility $\chi$ behaves according to eq. (24) with that of the equivalent tank impedance $Z_{\mathrm{eq}}$ of Fig. 3, one finds that the analytical and the circuit simulator model are equivalent if the following relations hold:

$$\frac{1}{\sqrt{L_{\mathrm{spin}}C_{\mathrm{spin}}}} = \omega_{\mathrm{L}} = -\gamma B_0 \tag{25a}$$

$$\frac{L_{\mathrm{spin}}}{R_{\mathrm{spin}}} = T_2/2 \tag{25b}$$

$$K_{\mathrm{spin}}^2 = \eta\chi_0, \tag{25c}$$

where it should be noted that according to the conventions used in this paper, $\gamma$ is a negative number. According to eq. (25), there are four parameters ($L_{\mathrm{spin}}, C_{\mathrm{spin}}, R_{\mathrm{spin}}$ and $K_{\mathrm{spin}}$, with $K_{\mathrm{spin}}$ being the coupling coefficient between the tank inductor of the VCO and the LC resonator modeling the spins, cf. Fig. 3), which model the spin ensemble in the circuit of Fig. 3 but only three parameters in the physical model without saturation ($\omega_{\mathrm{L}} = -\gamma B_0, T_2$ and $\eta\chi_0$), therefore one parameter can be chosen at will. Here, one natural choice could be to choose $L_{\mathrm{spin}} = L_{\mathrm{coil}}$, which always results in reasonable values for both $L_{\mathrm{spin}}$ and $C_{\mathrm{spin}}$.

## 8 Comparison between the analytical model and circuit simulations

In this section, we will compare the analytical signal and noise models of sections 4 and 5 against circuit simulations performed with Keysight's GoldenGate simulator. Accounting for the periodic nature of the solutions, the CR harmonic balance solver was used, defining the static magnetic field $B_0$ as a sweep variable to compute the field-sweep ESR spectrum numerically. To ensure a smooth transition between adjacent sweep points the corresponding flag was enabled in the CR analysis. The result of one such simulation is shown in Fig. 4. The corresponding simulation parameters are listed in the figure caption. These parameters correspond to the prototype realization discussed in section 9. According to the figure, there is excellent agreement between the analytical model and the circuit simulation. As highlighted by the arrows in the figure, there is a small asymmetry in the line shape, which reflects the fact that the amplitude ESR signal is both sensitive to the real part of the complex susceptibility, which displays a dispersive behavior, and the imaginary part of the susceptibility with its absorption characteristic. However, since the imaginary part is amplified by the (unloaded) coil quality factor (see eq. (8), $Q_{\mathrm{coil}} = 10.2$ for the simulation and the prototype of section 9)), the simulated ESR spectrum is mostly absorptive in nature with the small, but visible asymmetry introduced by the real part of $\chi$. Importantly, both the analytical model and the simulation accurately predict this behavior. The peak-to-peak amplitude of the spectrum is virtually unaffected by the real part of $\chi$, justifying the simplified expression of eq. (9), which was used to derive the LOD in section 6.

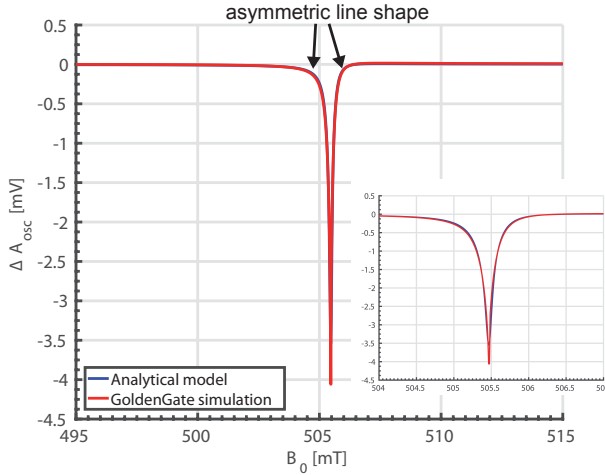

**Figure 4.** Comparison of the proposed analytical model for the amplitude variation in amplitude-sensitive ESR and GoldenGate simulations. The corresponding simulation parameters are: $f_{\mathrm{osc}}$ =14.209 GHz, $L_{\mathrm{coil}}$ =582.5 pH, $Q_{\mathrm{coil}} = 10.2$, $V_{\mathrm{TUNE}}$ =2.8 V, corresponding to $C\left(V_{\mathrm{TUNE}}\right)$ =103 fF. Loading of the previous stage was accounted for by load capacitors of $C_{\mathrm{L}}$ =75 fF to ground on both the positive and the negative oscillator output (AC coupled through 700 fF), transistor length $L = 120$ nm, transistor width $W = 12\,\mu$m, 24 fingers, technology: GFUS 130 nm CMOS, $I_{\mathrm{BIAS}}$ =1.75 mA, $\eta\chi_0 = 0.2 \times 10^{-3}\cdot 10^{-4}$, $T_2 = 60$ ns, $L_{\mathrm{spins}} = 100$ pH. $C_{\mathrm{spin}}$ and $R_{\mathrm{spin}}$ were automatically calculated for each sweep point from eq. (25). Furthermore, the condition for a non-saturated sample, i.e. $(\gamma\,B_1)^2 T_1 T_2 \ll 1$ was ensured. Inset: an enlarged view of the amplitude dip in both models.

In order to validate the analytical noise model of eq. (16), we have compared it against noise simulations performed using Keysight's GoldenGate simulator (CR analysis with noise enabled). Using the same simulation settings as for the simulations of Fig. 4 except for the bias current, which was varied as a parameter to analyze the range over which the proposed model is valid, we have obtained the results shown in Fig. 5. In the figure, the GoldenGate results are displayed as solid lines and the corresponding analytical model data are displayed in the same color with dashed lines. According to the figure, there is a good agreement between the proposed analytical model and the GoldenGate simulations in the white frequency noise region. Since Flicker noise was not taken into account in the model of section 5, the analytical curves start to deviate from the GoldenGate simulations for lower offset frequencies, where the Flicker noise produced in the cross-coupled transistor pair starts to dominate. The corner frequency at which the white noise floor starts to roll off is predicted by the analytical model within a factor of approximately two. For larger bias currents, the prediction of the white noise floor starts to deteriorate due to velocity saturation effects in the transistors, which are not taken into account in the simple square law model used to derive eq. (1).

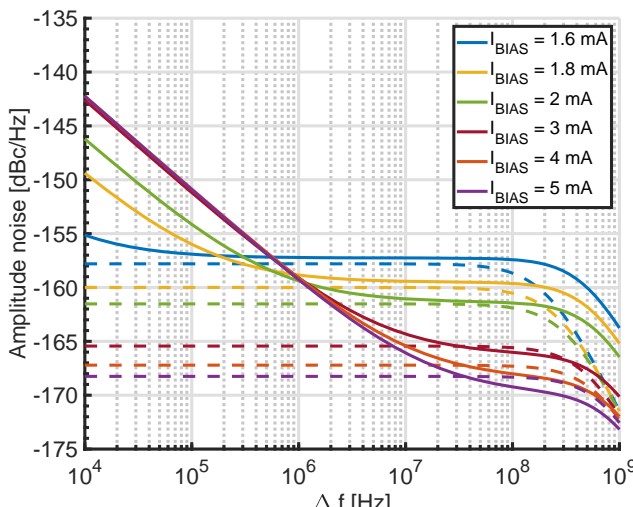

**Figure 5.** Comparison of the proposed analytical model for the amplitude noise of an LC tank VCO and GoldenGate simulations. The simulation parameters are identical to those listed in the caption of Fig. 4 except for the bias current $I_{\mathrm{BIAS}}$, which was varied as a parameter to evaluate the range over which the proposed model is valid. Solid lines correspond to the GoldenGate results and dashed lines to the analytical model.

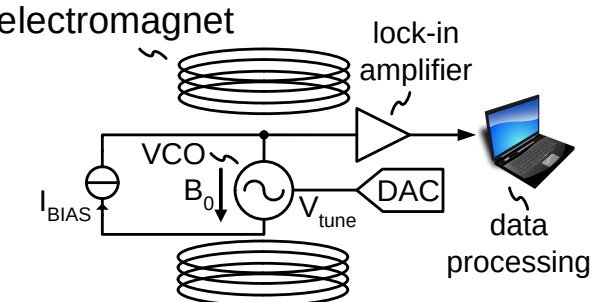

**Figure 6.** Experimental setup used to obtain the data of Fig. 7. The setup is almost identical to the one in Fig. 1b, with the difference of exchanging the permanent magnet for an electromagnet.

## 9 Measurements

In this section, we will compare the circuit simulator model of section 7 against measured data from a prototype realization of the proposed amplitude-sensitive VCO-based ESR sensor. To this end, we have used the ASIC, which was already presented by Handwerker et al. (2016), in the experimental setup of Fig. 6, using an off-chip low noise current source. Here, we have initially not used the setup of Fig. 1b, because the comparison between model and measured data turned out to be much simpler for field sweeps where the GoldenGate simulations take only a fraction of the time of frequency sweeps, where the oscillation frequency varies over a wide range with only very small ESR induced changes on top of these large electrical variations. Since

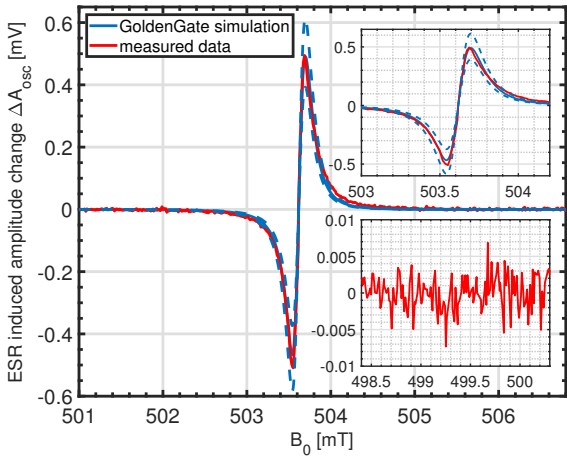

**Figure 7.** Comparison of the circuit simulator model of section 7 for the ESR induced amplitude shift with measured data of a 14 pl DPPH sample obtained using the VCO prototype presented in Handwerker et al. (2016) used in the setup of Fig. 1b. For comparison, the measured results are referred to the oscillator output by dividing them by the lock-in gain and the demodulation sensitivity of the built-in AM demodulator. Top inset: Zoom-in of the simulated and measured signals. Bottom inset: Zoom-in of the measured signal used for the noise calculation. Measurement conditions: $V_{\text{TUNE}} = 2.8\,\text{V}, I_{\text{BIAS}} = 1.7\,\text{mA}$, simulated ac current in coil $I_{\text{coil,peak}} \approx 8\,\text{mA}$ corresponding to $B_1 \approx 16.8\,\mu\text{T}$.

there is, in principle, a one-to-one correspondence between field and frequency sweeps and we use $V_{\text{TUNE}}$ to introduce a frequency modulation via a DAC, i.e. additional noise via the tuning port is considered in the experiments, these field sweeps display the same SNRs but make the comparison with the model much simpler. This being said, we also validated the proposed simplified setup of Fig. 1b experimentally.

Since according to the simulation results of Fig. 5, the noise power spectral density around $\Delta f = 0$ is heavily plagued by 1/f-noise[3], we have introduced a lock-in detection scheme (off-chip lock-in amplifier), by modulating the oscillation frequency at a frequency of $f_{\text{mod}} = 10\,\text{kHz}$ using a sine wave voltage applied to the VCO tuning voltage $V_{\text{TUNE}}$. In this way, we have measured the ESR spectrum of a small DPPH (2,2-diphenyl-1-picrylhydrazyl, Sigma Aldrich) sample with a volume of approximately 14 pl shown in red in Fig. 7. The solid blue line in Fig.7 corresponds to a GoldenGate simulation using a filling factor corresponding precisely to the estimated sample volume of 14 pl and the two dashed blue lines indicate two additional simulations with filling factors corresponding to an error in the estimation of the sample volume of $\pm 25\,\%$. Here, we have taken into account the demodulation sensitivity from the oscillator output voltage to node $\boxed{\text{X}}$ in Fig. 2a (simulated to be 1/7.9 V/V) and the simulated lock-in detection spectra were computed from the direct detection spectra, cf. Fig. 4, in MATLAB using the same modulation amplitude (frequency modulation with an amplitude of $1.5\,\text{mV}_{\text{rms}}$, which together with the VCO slope of 0.8 GHz/V corresponds to an equivalent peak-to-peak field modulation of $120\,\mu\text{T}$) that was used in the measurement. Accord-

---

[3]parts of the spectrums with a slope of $-10\,\text{dB/dec}$

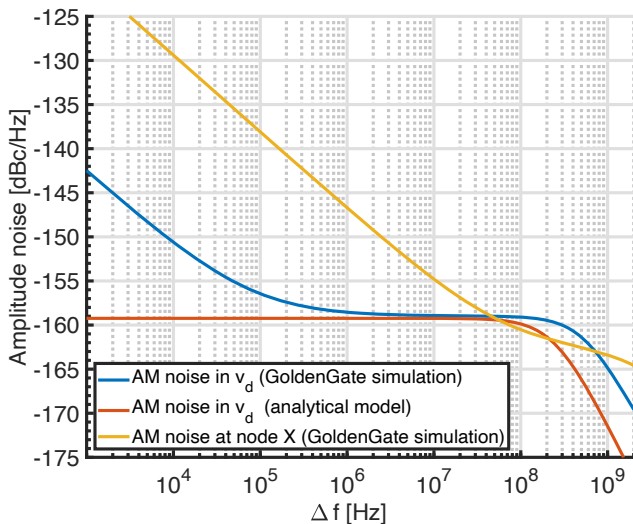

**Figure 8.** Comparison of the GoldenGate simulations of the amplitude noise in the VCO output voltage $v_{\mathrm{d}}$ and the analytical model of section 5 and with the simulated noise floor at the demodulated output labeled $\mathrm{VDD_{osc}}$ in Fig. 2.

ing to the figure, when taking into account the modeling uncertainty due to difficulties in precisely determining the sample volume, there is an excellent agreement between the proposed circuit simulator model and the measured data.

To estimate the spin sensitivity of our system, we have used the measured data presented in Fig. 7. With the calculation detailed in Appendix A, we have estimated a spin sensitivity of approximately $N_{\min} = 8.9 \times 10^{10}\,\mathrm{spins}/(\mathrm{G} \cdot \sqrt{\mathrm{Hz}})$, which is 445 times worse than the theoretically predicted value of $N_{\min} = 2 \times 10^8\,\mathrm{spins}/(\mathrm{G} \cdot \sqrt{\mathrm{Hz}})$. This is in part due to the $B_1$ used in the measurements $B_{1,\mathrm{meas}} \approx 16.8\,\mu\mathrm{T}$ being approximately 8 times less than $B_{1,\mathrm{opt}} \approx 140\,\mu\mathrm{T}$ to avoid any line broadening, leading to a reduction of approximately 2.5x in $N_{\min}$, according to eq. (18). Here $B_{1,\mathrm{opt}}$ is calculated from eq. (19) using $T_1 = T_2 \approx 41\,\mathrm{ns}$ as extracted from the measured peak-to-peak linewidth of 1.4 G (cf. Fig. 7, equivalent to an FWHM of about 3 G), while $I_{\mathrm{BIAS}}$ and $\alpha_{\mathrm{od}}$ used in the measurement and in the optimum SNR condition are extracted from simulations, to be respectively 1.7mA, 1.25 and 27.2mA, 5. This large $\alpha_{\mathrm{od,opt}}$ of 5 leads to a higher amplitude noise in the optimum SNR condition (cf. eq. (16)), explaining the $N_{\min}$ reduction of 2.5x mentioned above. To investigate the remaining discrepancy of $445/2.5 \approx 180$, we have also simulated the amplitude noise at node $\boxed{\text{X}}$ in Fig. 2a, which is depicted together with the simulated AM noise in the oscillator output voltage and the analytical noise model in Fig. 8. According to the figure, although the demodulation sensitivity between the oscillator output voltage and node $\boxed{\text{X}}$ is less than unity, the noise floor is significantly larger leading to a greatly reduced SNR in the demodulated output. More quantitatively, the demodulation sensitivity of $1/7.9\,\mathrm{V/V}$ together with the 30 dB increased noise floor (an 8 dB increase due to the 1/f-noise, which is not accounted for in eq. (23), and another 22 dB increase in the demodulated output at node $\boxed{\text{X}}$) predict an factor of 250 difference between the theoretically predicted value of $N_{\min}$ and the measured value. This agrees well with the discrepancy of 180 mentioned above. Moreover, the corner frequency between the 1/f- and the white noise parts of the spectrum occurs at significantly larger frequencies, effec-

325 tively preventing an operating in the white noise region because at such high modulation rates rapid scan effects that perturb the spectra (Tseitlin et al., 2011) would already become visible. To verify the accuracy of the GoldenGate noise simulations, we have used the simulated noise floor at node $\boxed{X}$ of -129.5 dBc/Hz and the simulated oscillator amplitude of approximately $820\,\mathrm{mV}$ to predict the rms noise in the measured data of Fig. 7. Taking into account the lock-in gain of 100 and the lock-in bandwidth of 2.5 Hz, the simulated noise floor predicts an rms noise of $3.5\,\mu V_{\mathrm{rms}}$, which corresponds approximately to the

330 measured rms noise of $2.3\,\mu V_{\mathrm{rms}}$[4].

Finally, we have also performed frequency scan ESR experiments where instead of sweeping the static magnetic field $B_0$, the tuning voltage $V_{\mathrm{TUNE}}$ is ramped in and out of resonance. An example spectrum of a DPPH sample with a volume of approximately 3 pl obtained using this method is shown in Fig. 9. In these frequency sweep experiments, we have achieved the same sensitivity as in the field sweep experiments. Similarly to previous experiments, the VCO gain was 0.8 GHz/V and

335 a noise floor of $\approx 0.1\,\mathrm{mV_{rms}}$ can be observed, in accordance with our models. Performing frequency sweeps allows for the use of the simplified experimental setup of Fig. 1b, which is ideally suited for future point-of-care ESR spectrometers, without performance degradation. This being said, care has to be taken that no additional noise is introduced into the system via the tuning voltage input $V_{\mathrm{TUNE}}$. At this point, it is convenient that the amplitude-sensitive detection setup is much more immune against such additional noise from $V_{\mathrm{TUNE}}$ than the frequency-sensitive setup, where the well-known AM-to-PM conversion in the varactor makes the setup much more prone to an increase in the noise floor due to DAC noise in $V_{\mathrm{TUNE}}$.

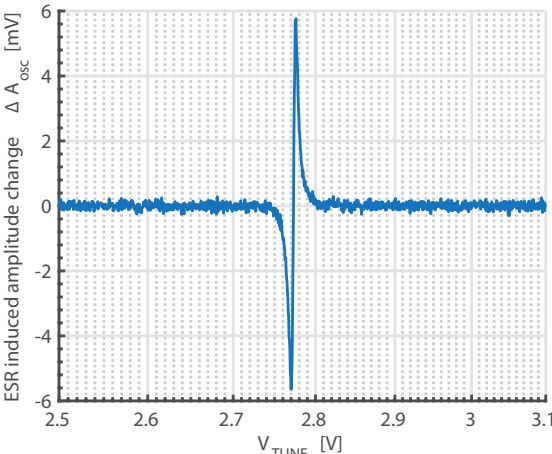

**Figure 9.** Measured spectrum of a DPPH sample of a volume of approximately 3 pl. The spectrum was acquired using the setup of Fig. 1b by sweeping the tuning voltage of the VCO through the resonance frequency and simultaneously applying a small sinusoidal signal for a subsequent lock-in detection.

---

[4]Here, it should be noted that the spectrum of Fig. 7 is referred back to the oscillator output, i.e. divided by the lock-in gain and the demodulation sensitivity.

## 10 Conclusions

In this paper, we have introduced a new ESR detection method, which senses the ESR effect as changes of the amplitude of an integrated VCO. Together with the proposed setup, we have presented analytical models for both the ESR-induced amplitude changes and the AM noise floor of the VCO-based detector. The analytical models were then used to predict the limit of detection of the proposed method, which was shown to be identical to that of the previously presented frequency-sensitive VCO-detection approach presented by Handwerker et al. (2016). The analytical models were then verified against circuit simulations including an RLC tank model for the ESR effect. Finally, we have validated the circuit simulator model against measured data obtained from a VCO prototype operating around $14\,\mathrm{GHz}$. When taking into account the increased noise floor at the intrinsic AM demodulation point inside the VCO, we have achieved a very good agreement between model and measured data, clearly showing that the circuit model can be used to optimize the detector performance already early on in the design phase, thereby removing the need for costly and time-consuming hardware iterations. Moreover, the presented results serve as proof of principle that with the proposed approach good sensitivities can be achieved already at moderate ESR frequencies. Since the proposed method scales very advantageously with frequency, it can fully benefit from the current ESR trend of going to higher and higher fields (and therefore also operating frequencies) to further improve sensitivity. With its very simple experimental setup, cf. Fig. 1b, and the availability of permanent magnets with field strengths up to approximately $2\,\mathrm{T}$, the proposed approach is ideally suited for the design of future generation of portable ESR spectrometers, which can play a crucial role in emerging fields such as on-site food quality control, manufacturing process control or potentially personalized medicine and home diagnostics.

## Appendix A: Spin sensitivity calculation

We used the measured data shown in Fig. 7 to calculate the spin sensitivity. For the mass density of DPPH, we used the mean value of densities for the various DPPH crystal forms reported by Kiers et al. (1976); Williams (1967); Wang et al. (1991b, a) as $1.4\,\mathrm{g/cm^3}$. As pointed out by Matsumoto and Itoh (2018), the number of radicals per unit mass can vary between different manufacturers due to the different purities and solvents used up to almost 20%, with a mean value from their 3 samples of approximately $1.4 \times 10^{21}\,\mathrm{spins/g}$. Combining the two numbers, we arrived at a spin density of approximately $2 \times 10^{12}\,\mathrm{spins/pL}$. Our DPPH sample has a volume of approximately $14\,\mathrm{pL}$, resulting in a total number of $2.8 \times 10^{13}$ spins. From the measurement data, the rms noise floor is calculated from the first 140 samples (the bottom inset in Fig. 7) to be $2.35\,\mathrm{\mu V}$, while the signal amplitude is approximately $1\,\mathrm{mV}$, leading to an SNR of 426. Since we used a lock-in BW of $2.5\,\mathrm{Hz}$, the SNR per unit measurement time is $673.6\,\sqrt{\mathrm{Hz}}$. Using this value together with the above-mentioned number of spin, the measured DPPH linewidth of $1.4\,\mathrm{G}$, and eq. (22), we calculated the spin sensitivity to be $8.9 \times 10^{10}\,\mathrm{spins/(G \cdot \sqrt{Hz})}$.

*Author contributions.* AC designed the chip, ran simulations, designed the measurement setup and conducted measurements. BS designed the measurement setup and conducted measurements. MK ran simulations, designed the measurement setup and conducted measurements.

JLG designed the measurement setup and conducted measurements. AA and KL conceived the experiments. JA conceived the idea and experiments, derived the analytical models, designed the measurement setup and conducted experiments. All authors contributed to the manuscript.

*Competing interests.*   No competing interests declared.

*Acknowledgements.*   This work was supported by DFG projects AN 984/16-1, AN 984/8-1, and Bundesministerium für Bildung und Forschung under contract number 01186916/1 (EPRoC). A. Angerhofer acknowledges support through the DSR Opportunity Fund from the University of Florida. The authors gratefully acknowledge fruitful discussions with Klaus Peter Dinse and Silvio Künstner.

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
