# Peer review of "On the modeling of amplitude-sensitive ESR detection using VCO-based ESR-on-a-chip detectors"

_Magnetic Resonance, 2021_

## Author Comment (AC1)

**On the modeling of amplitude-sensitive ESR detection using VCO-based ESR-on-a-chip detectors**

**Reply from the authors**

Dear Editor, dear reviewers,

We would like to thank you for considering our manuscript for publication in Magnetic Resonance and for the overall positive evaluation of our work. Moreover, we would like to thank the reviewers for their extremely useful remarks. Their detailed and very insightful comments were instrumental in further improving the quality of our manuscript. Please find our detailed responses to the individual comments below.

With kind regards,
Michal Kern
On behalf of all authors

**RC1**

*General comments: The authors present a variation on a theme for their VCO-based ESR detection systems, switching from oscillator frequency to amplitude-based detection of the ESR signal. The manuscript is not written for the general practitioner of EPR spectroscopy; however, documenting and validating the authors' approach in specific sections with relevant equations and assumptions provides important transparency for specialized groups. It would be advisable to tone-down the emphasis on point-of-care EPR spectrometers unless the authors can point to specific medical tests (either FDA/EMA or CLIA assays) that use EPR spectroscopy for diagnostics and/or clinical decision-making. Although my expertise does not extend to review/evaluation of the mathematics and engineering aspects of the manuscript, the work is a meaningful contribution for those developing portable ESR spectrometer systems as the amplitude-detection mode simplifies the experimental setup and does not appear to deteriorate performance (but see notes below on the spin sensitivity calculations).*

We thank the reviewer for their in-depth and overall positive review of our manuscript.

**Specific comments:**

*Page 1, line 19: True, but I prefer the introduction to EPR presented in Schlecher et al. IEEE SENSORS JOURNAL, VOL. 19, NO. 20, OCTOBER 15, 2019 because it pitches EPR more positively. (EPR seems to be a shrinking community...)*

We fully agree with the reviewer and will modify the introduction accordingly.

*Page 2, line 22: Additional recent references: Dayan et al. Rev. Sci. Instrum. 89, 124707 (2018); https://doi.org/10.1063/1.5063367; Abhyankar et al. Sci. Adv. 28 Oct 2020: Vol. 6, no. 44, eabb0620 https://doi.org/10.1126/sciadv.abb0620*

Thank you for bringing these additional references to our attention. We will add these references to the revised version of the manuscript.

*Page 2, line 32: As far as I am aware, there are no EPR spectroscopy-based FDA/EMA approved tests for any diseases/conditions, making the need for such instrumentation extremely aspirational (and not based on documented need). Portable EPR spectrometers are likely most useful for point-of-production food analysis (beer, wine, olive oil) and oil/gas analysis.*

We agree with the reviewer that there are no approved EPR-based tests. We know that there is strong interest from the industry (anecdotally, company Noxygen) and even a documented need in literature (Romanyukha et al 2014, Cristea et al 2020). Nevertheless, we will tone down the need for PoC EPR in medicine and add references for food analysis.

*Page 3, line 64: To help those who do EPR spectroscopy, but who aren't EE's: I believe this description is how the "resonator dip" is displayed on commercial systems for tuning purposes. A voltage ramp is applied to a VCO to produce a "frequency window", the center of which is varied so the user can identify the resonator "dip".*

We thank the reviewer for bringing this explanation in our manuscript to our attention which apparently is not written in a sufficiently clear manner. In the VCO-based approach, the frequency ramp can be conveniently generated simply by applying a ramp signal to the VCO tuning voltage. This voltage both defines the new frequency value and implicitly tunes the LC tank inside the VCO to the correct, on-resonance value. This is because, in a VCO, the oscillation frequency and the resonance frequency of the LC tank are identical. This is in contrast to a conventional resonator-based scheme, in which the resonance frequency and the excitation frequency can be independently defined. We will add a few lines to the revised manuscript to explain this important difference in more detail.

*Figure 3, page 10: This picture is also shown in https://ieeexplore.ieee.org/document/8310330. Some elements of the AM detection also are presented there: solid DPPH and solid TEMPOL on two different array coils (Figure 21.6.5, right side).*

We would like to thank the reviewer for bringing this missing reference to our attention. We have indeed presented preliminary results in the AM detection mode in the 1-page conference abstract in https://ieeexplore.ieee.org/document/8310330 . In the revised version of the manuscript, we will properly cite this preliminary work and explain the modeling advances that we have accomplished since then in detail.

*Page 12, line 281: "Since according to the simulation results of Fig. 5, the noise power spectral density around Δω= 0 is heavily plagued by 1/f-noise,…" Not being an EE, it was not clear to me how Figure 5 showed this problem.*

Thank you for bringing this unclear statement in the first version of the manuscript to our attention. In the revised version of the manuscript, we will correct the x-axis label to $\Delta f = \Delta\omega/2\pi$, clearly indicate the 1/f noise (parts of the spectrum with a slope of -10 dB/dec) and the white noise (flat power spectral density) parts of the spectra and adapt the wording using less EE jargon.

*Figure 7, page 14: The DPPH linewidth in Figure 7 is about 0.5 mT (5 G), which is a bit larger than I would expect for solid DPPH detected using conventional EPR spectroscopy (2*

*G, see Yalcin & Boero 2008).  An assumption was made that only the imaginary part (absorption) of the magnetic susceptibility is important in this detection scheme (see text before equation 9).  Can the authors explain why the linewidth for solid DPPH at room temperature here is broader than expected by more than a factor of 2?*

We thank the reviewer for their careful examination of the presented data. This comment has prompted us to perform additional experiments and simulations. As an outcome of these experiments, we can indeed confirm that the data previously presented was subject to overmodulation. We have now corrected this and present a new dataset with a linewidth of 1.4 G.

*Page 15, lines 295 & 296:  The Yalcin & Boero 2008 reference lists N of $2 \times 10^{27}$ spins/$m^3$ for DPPH in the caption of Figure 4 and again in the text.  There is no reference to how it was calculated.  It would be helpful to refer people to one of the original DPPH crystal structure papers (https://cdnsciencepub.com/doi/pdf/10.1139/v91-194  or https://link.springer.com/content/pdf/10.1007/BF01066204.pdf ).  Note that in these papers, the space group differs and therefore the #spins/volume varies by about 20 %.*

We agree that the original references should be cited and regret the omission. The first reference relates to a 2,2-di(p-nitrophenyl)-1-picrylhydrazine dichloromethane complex and the second one to the diamagnetic 2,2-diphenyl-1-picrylhydrazine. While both are related to 2,2-diphenyl-1-picrylhydrazil, commonly referred to as DPPH in EPR literature, they are different. The crystal structure can indeed vary in DPPH based on the solvent used (there are at least four forms of DPPH, referred to as DPPH(I) to DPPH(IIIa), with crystal structures reported for three of them). However the reported differences in unit cells are smaller than 10% (Kiers et al 1976, Williams 1965), with an approximate mean value of reported mass densities of the various crystal forms of 1.4 g/$cm^3$.

However, as pointed out by Matsumoto & Itoh (2018), the number of radicals per unit mass can vary between different manufacturers due to the different purities and solvents used up to almost 20%, with a mean value from their 3 samples of approximately $1.4 \times 10^{21}$ spins/g.  In order to avoid potentially compounding the same uncertainty twice, we will use the above-mentioned mass and spin densities, with a 20% uncertainty of the latter for calculations of the spin sensitivity in the revised version of the manuscript.

*Calculation of $N_{min}$:  Using either 1 spin/$4.84 \times 10^{-13}$ pL or 1 spin/$5.85 \times 10^{-13}$ pL (see references cited above for these spin/unit cell volume values; 1 $Å^3 = 1 \times 10^{-15}$ pL) and a 23 pL volume the authors give for the DPPH sample size, the #spins in the sample volume is ($4.3 \pm 0.4) \times 10^{13}$.  In eq. 21, assuming an optimal SNR of 3 (typical in analytical chemistry) cancels the 3 in the numerator, so that $N_{min}$ becomes $N_{spins}$.  Dividing $4 \times 10^{13}$ by the 5 G linewidth measured here, gives $2 \times 10^{12}$ spins/G sqrt(Hz).  Using eq. 11 in Yalcin & Boero 2008, $N_{min} = (1/SNR)(N_{spins} * V_{sample}/sqrt(f_{BW}))$ gives $1.4 \times 10^{13}$ spins/sqrt Hz.  Dividing that by 5 G, gives approximately $3 \times 10^{12}$ spins/G sqrt Hz.  These two values are very close to one another.  From Figure 7, the signal intensity is 1.2 mV and the baseline noise (estimated) is 0.02 mV, which gives an SNR of 60; using that SNR gives an $N_{min}$ of $1.4 \times 10^{11}$ spins/G sqrt Hz.  My question: how did the authors arrive at $N_{min} = 2 \times 10^{10}$ spins/G sqrt Hz using equation 21?*

Thank you very much for bringing this inconsistency to our attention. Triggered by the reviewer's comment, we have reexamined our calculation with the newly recorded, nonovermodulated dataset mentioned above. With this new dataset we calculate a spin sensitivity of $N_{min} = 1.3 \times 10^{11}$ spins/G sqrt Hz. For the new experiments, we have used a sample with an approximate volume of 14 pL, a measurement bandwidth of 2.5 Hz. With a measured SNR of approximately 433 with a linewidth of 1.4 G. Using the above-mentioned spin density of $1.4 \times 10^{21}$ spins/g, mass density of 1.4 g/cm$^3$, we arrive at approximately $2 \times 10^{12}$ spins/pL and consequently, $2.7 \times 10^{13}$ total spins in the sample. By using these numbers in the same equations, we arrive at approximately $N_{min} = 1.3 \times 10^{11}$ spins/G sqrt Hz. The discrepancy between the new and previous numbers mostly arises from an error in our measurement script that wrongly translated from the lock-in time constant to the effective noise bandwidth, as well as the overmodulation. To make the spin sensitivity calculation fully traceable, we will include all details, including a zoomed-in version of the noise floor in the revised version of the manuscript.

*Page 15, lines 313-314: Looking at Figure 9, I guesstimated that the linewidth is about 5 mV. Using the VCO gain 0.8 GHz/V and 28 GHz/T gives 1.4 G for the DPPH linewidth in Figure 9, which is closer to the typical 2 G linewidth for DPPH.*

Thank you for bringing this inconsistency to our attention. As mentioned above, the data in Fig. 7 were overmodulated, explaining the different linewidths.

*Page 16, line 335: Is there a documented need for portable EPR spectrometers for personalized medicine? This claim appears to be strongly investigator-driven and not market-driven.*

We agree with the reviewer that this claim is mostly investigator-driven, and we will therefore tone it down in the revised version of the manuscript.

**Technical Corrections:** Generally, the written English is excellent. There are few suggestions/corrections below.

*Page 2, line 54: sweeping" and "static" are a bit counter to one another. The field is not swept through its resonance (implied with the current wording); it is swept to achieve resonance with the energy splittings in the sample. Although I understand what is meant, perhaps making it two sentences would be useful. "An ESR experiment...reflected power. The externally-applied magnetic field $B_0$ is swept through the resonance condition where the sample's energy level splittings match the applied frequency.*

*Page 4, line 75: This sentence, as written, implies that the "very simple experimental setup" is from Handwerker et al. Perhaps revise as follows? "This hardware change simplifies the experimental setup compared to the frequency-sensitive detection described by Handwerker et al. (2016). In that report, the VCO output signal first had to be processed by a chain of frequency dividers to allow for simplified analog-to-digital conversion and subsequent frequency demodulation by a digital phase-locked loop. Fig. 1b shows that such additional elements are not required in the current implementation."*

*Page 7, line 170: Reference needs parentheses to enclose it.*

*Page 11, line 247: "defining" should be "to define"*

*Page 11, line 250: "there is an excellent agreement" should be "there is excellent agreement"*

*Page 15, line 343: "where" should be "were"*

We thank the reviewer for their careful reading of the manuscript and will correct all of these errors in the revised version of the manuscript.

**RC2**

**General comments:**

*The paper presents a detailed analysis of the amplitude sensitive ESR detection using VCO based detector. In this method the amplitude of the oscillator is monitored to detect the ESR signal unlike the common method where the change in frequency is used to detect the ESR signal. Similar analysis was done for oscillator frequency change detection method, previously reported by the authors. Though the method is not new, a detailed analysis of the setup using circuit simulations and the comparison of simulated and measured outcome is noteworthy. It is also important to note that the amplitude sensitive detection allows for simplified signal detection setup compared to the frequency detection setup especially when the frequency modulation may be quired for enhance the signal to noise to ratio.*

*The previous work on the amplitude sensitive detection by Matheoud is mentioned by the authors but a simple comparison of the previous work and the work detailed in this paper will have some value (like SNR comparison and mention of HEMT and CMOS technologies).*

We thank the reviewer for their overall positive assessment of our work. We will include the requested comparison to previous work on amplitude-sensitive EPR using oscillators in the revised version of the manuscript.

**Specific comments:**

*Page 4, line 90 : define slope factor ( something like transistor characteristic dependent parameter)*

We will change the slope factor to "transistor slope factor", specify the typical numerical value of n = 1.3, and add a reference to relevant literature.

*Page 5, line 98 : vd and Aosc,0 is not defined. Defining this here would help understanding eq 8 better where Aosc,0 is when there is no ESR.*

$v_d$ is the differential voltage across the LC tank, as defined in Figure 2 and Page 5, line 91. Aosc,0 is defined in eq. (3a). Given the confusion that might have arisen from the double definition in eq. (3a) $\hat{v}_d = A_{osc,0}$, we will consider simplifying the revised version of the manuscript.

*Page 6, line 121 : remove "also" from " also on the coil inductance….."*
We will correct this mistake in the revised version of the manuscript.

*Page 6, line 131 : see Page 5, line 98 comment*

Please refer to our answer to the comment above.

Page 6, line 142 : remove "away" from " system far away from"
We will correct this in the revised version of the manuscript.

*Page 8, line 196 : The equation is missing ωosc on the right hand side.*
We will correct this in the revised version of the manuscript.

*Page 9, line 203 : insert "with respect to B1 and Wosc" after "derivatives of eq. (17)*
We will correct this in the revised version of the manuscript.

*Page 9, line 204 : insert " respectively" after " oscillation frequency wosc.."*
We will correct this in the revised version of the manuscript.

*Page 9, line 205 : mention of eq 4(b) to obtain eq 18 will be useful*
Thank you for pointing this out. We will include this additional reference to eq. (4b) in the revised version of the manuscript.

*Page 9, line 213 : line " where Nspins is the number of spins in the sample and SNRopt is the optimum SNR achievable with said number of spins for" is awkward maybe just "where Nspins is the number of spins in the sample for SNRopt" is better ???*
We will revise this sentence in the updated version of the manuscript.

*Page 10, line 236 : mentioning or equating eq. 24a to frequency for ESR would be helpful to connect the dots between the circuits model and analytical model*
Thank you for pointing this out. We will include this additional reference to eq. (24a) in the revised version of the manuscript.

*Page 10, line 238 : define K (coefficient of coupling)*
Thank you for pointing this out. We will include this missing definition in the revised version of the manuscript.

*Page 11, line 241 : remove and before " therefore one parameter can be"*
We will correct this in the revised version of the manuscript.
*Page 11, line 246 : what is CR analysis is it a function in the simulator?*
CR Analysis is the trademark name for the Harmonic Balance solver in the GoldenGate simulation software offered by Keysight. We will change the revised manuscript by simply referring to a harmonic balance solver to avoid confusion.

*Page 11, line 253 : replace "said" with "the" in " part of said susceptibility"*
We will revise this sentence in the updated version of the manuscript.

*Page 11, line 254 : mentioning section 4 in this line would help the reader go back to the section to understand the effect of Qcoil on the imaginary part of the susceptibility*
Thank you for pointing this out. We will include this additional reference to section 4 in the revised version of the manuscript.

*Page 15, line 323 : replace "where" with "were" in "The analytical models were then used…"*
We will correct this in the revised version of the manuscript.

**References**

Cristea, D., Wolfson, H., Ahmad, R., Twig, Y., Kuppusamy, P., & Blank, A. (2021). Compact electron spin resonance skin oximeter: Properties and initial clinical results. *Magnetic Resonance in Medicine*, *85*(5), 2915-2925. https://doi.org/10.1002/mrm.28595

Kiers, C. T., De Boer, J. L., Olthof, R., & Spek, A. L. (1976). The crystal structure of a 2,2-diphenyl-1-picrylhydrazyl (DPPH) modification. *Acta Crystallographica Section B*, *32*(8), 2297-2305. https://doi.org/https://doi.org/10.1107/S0567740876007632

Matsumoto, N., & Itoh, N. (2018). Measuring Number of Free Radicals and Evaluating the Purity of Di(phenyl)-(2,4,6-trinitrophenyl)iminoazanium [DPPH] Reagents by Effective Magnetic Moment Method. *Analytical Sciences*, *34*(8), 965-971. https://doi.org/10.2116/analsci.18p120

Romanyukha, A., Trompier, F., Reyes, R. A., Christensen, D. M., Iddins, C. J., & Sugarman, S. L. (2014). Electron paramagnetic resonance radiation dose assessment in fingernails of the victim exposed to high dose as result of an accident. *Radiation and Environmental Biophysics*, *53*(4), 755-762. https://doi.org/10.1007/s00411-014-0553-6

Williams, D. (1965). Crystallographic Data for 2,2-diphenyl-1-picrylhydrazyl. *Journal of the Chemical Society (London)*.